# The Use of Precise Survey Techniques to Find the Connection between Discontinuities and Surface Morphologic Features in the Laže Quarry in Slovenia

**Aleš Lazar [1],\*** , **Goran Vižintin [2]** , **Tomaž Beguš [3] and Milivoj Vulić [2]**

[1]   Geoservis, d.o.o., 1000 Ljubljana, Slovenia
[2]   Faculty of Natural Sciences and Engineering, University of Ljubljana, 1000 Ljubljana, Slovenia;
     goran.vizintin@guest.arnes.si (G.V.); milivoj.vulic@guest.arnes.si (M.V.)
[3]   Geotrias d.o.o., 1000 Ljubljana, Slovenia; t.begus@gmail.com
\*   Correspondence: ales.lazar@geoservis.si; Tel.: +386-01-5863830

**Abstract:** This paper addresses a stability evaluation of artificial slopes in a quarry located in Slovenia that was affected by a rockslide in March 2019. In order to ensure the safety of further production, measures were taken to restore the slopes. A stability assessment of the remaining parts of the quarry was conducted. To ensure quality spatial data, an upgraded study based on terrain mapping and aerial photogrammetric imaging using an unmanned aircraft was carried out, in addition to a traditional field survey of the quarry. So that the data were qualitatively useful, a digital terrain and discontinuity model was developed. Projections of the discontinuities occurring in the quarry and in the wider area were determined. The focus of the modeling was finding the main systems of discontinuities and projecting these systems onto the unexcavated parts of the quarry.

**Keywords:** discontinuity; stability analysis; geomorphological analysis; sliding surface; quarry; remote sensing

---

## 1. Introduction

In March 2019, a rockslide involving several benches occurred in the Laže 1 quarry near the village of Laže in Slovenia. The volume of the sliding mass was estimated to be 45,300 m$^3$. The rockslide diminished production in the quarry (see Figure 1). Then, a problem arose when the stability of the remaining areas of the quarry was questioned. To prepare for the restoration of the slope, the parameters of the rock mass were determined to design the measures needed for further optimal excavation and monitoring of the (in)stability of the slopes.

The work was carried out in several stages. First, the existing documentation was reviewed. Then, a geological field survey of the quarry was carried out. This was followed by upgrades based on terrain mapping and unmanned aircraft data: after each stage, the preliminary results were reviewed, and the basic model of the quarry area was updated. For the qualitative usefulness of the data, a digital terrain and discontinuity model was developed. To prepare for restoration, geological engineers, quarry operators, and design engineers communicated with each other often.

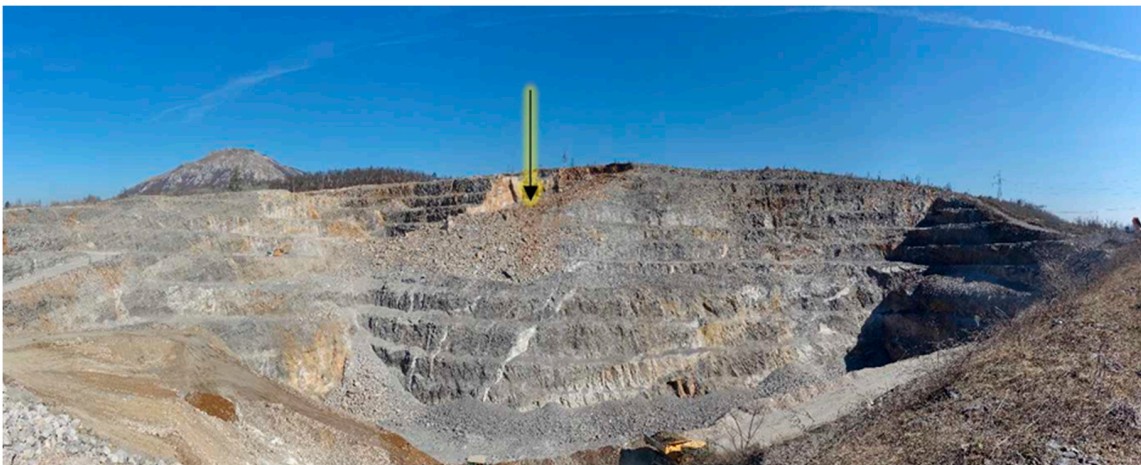

**Figure 1.** View toward the north. The collapsed area is marked with an arrow.

Risk analysis is an essential part of preventive action and a cornerstone of spatial plan assessments and other programs [1,2]. The first prerequisite for successful prevention is to recognize and understand the properties of slope mass movements and the local factors that cause them [3]. The properties of these processes can be determined by establishing a monitoring system that performs periodic and systematic observations and obtains data on changes in time and space. For this purpose, different measurement methods are often used [4–12].

Technological developments in unmanned aerial vehicles (UAVs) and data processing technology have led to the extensive use of these techniques in various fields [13–19]. Remotely piloted systems are able to support numerous mapping applications [20–27]. UAV techniques can detect rapid changes in morphology and topography [19,28–34]. UAVs have also been used for the environmental modeling of open-pit gravel mines and to measure the change in volumetrics [19,35]. This technology effectively supports geological and geotechnical engineering applications [36]. One of the most common applications is an assessment of slope stability and geohazards [19,37–40]. Gathering data for a stability assessment includes imaging slope profiles and block sizes and detecting possible discontinuity surfaces that can contribute to slope instability [41].

Previous work has demonstrated that using UAVs is a feasible and rapid method to obtain slope geometry data and morphological changes [40,42,43]. Slope profiles can be obtained from various point cloud sources [44,45]. Remote sensing techniques are capable of identifying structural geological elements [38,46]. Novel methods and close-range surveys using digital techniques include the use of a thermo-camera [47], close-range photography [43,48], and Light Detection And Ranging (LIDAR) [49,50]. However, at many of these sites, steep cliffs mean limited access, and onsite measurements are partly or entirely impossible due to potential rockfalls. This is where remote sensing techniques can be useful. These tools are often applied in assessments of structural geological conditions [40,49] or to indicate the location of faults [46,51] or folds [52]. Several studies have demonstrated that UAV-based surveys can be especially beneficial to the mining industry, especially open-pit mining [19,53]. In many previous geological engineering studies, remote sensing techniques have focused on slope stability and rockfall hazard assessments by surveying the area of interest with UAVs [19,37,38,54–58].

The aim of this research was to find the connection between discontinuities in the quarry and the surface morphologic features of the entire study area. During the fieldwork, tools that could quantitatively estimate rock parameters were used. A terrain model was produced by studying available data from state recordings [59] (orthophoto data, topographic plans M 5000, 25,000), LIDAR imagery from state networks, up-to-date geodetic survey data of the terrain, and images from our own unmanned aerial vehicle (UAV). During the research period, we used geological mapping data from the quarry and analyzed the spread of these elements in a terrain model. Slope stability calculations

were also performed, and we determined the projection of cracks occurring in the quarry and over the surrounding area.

### 1.1. The Study Area

The Laže 1 quarry is located just 1 km north of the village of Laže in the municipality of Postojna. It was built on a slightly spherical plateau where sinkholes had appeared. There is no running water in the wider area.

The original mining project established a general slope of 38° and a bench slope angle of 70°. The quarry runs 400 m in the E–W direction and 300 m in the N–S direction. The height difference between the highest and lowest points is 110 m. Excavations are carried out using benches that are 10 m apart. The bottom bench is at a height of 627 m a.s.l., and the top bench is at 730 m a.s.l.

On 13 March 2019, a rockslide occurred during blasting operations when 1000 kg of explosive material was used 20 m away from where the rockslide began (see Figure 2). No falling material was observed previous to this.

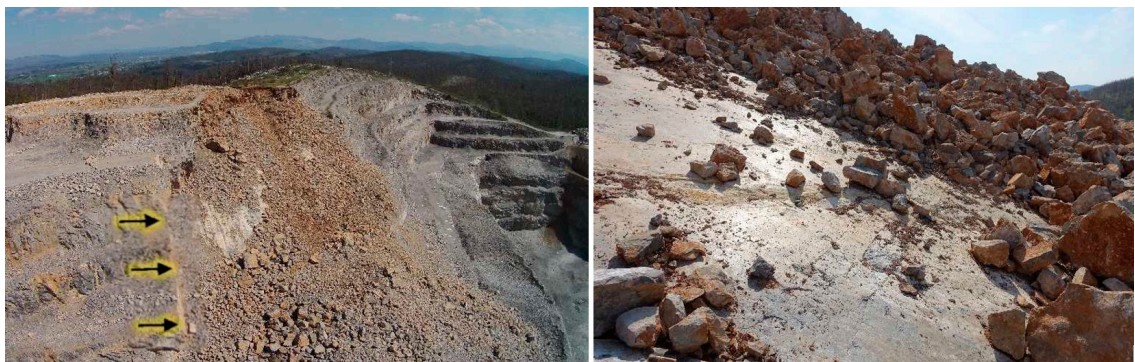

**Figure 2.** Left: view of the rockslide toward the southeast. A fault plane is clearly visible. Vertical cracks on the west side are noted with arrows. Right: the flat sliding surface in the rockslide area, facing east.

### 1.2. Geological Setting of the Area

The Official Geological Map of Slovenia (the Postojna section) [60] indicates that the main material in the quarry is limestone. According to this map, the area where the Laže 1 quarry is located consists almost entirely of Danian and Kozinian limestone (K, Pc), except for a narrow belt in the southwestern part, which consists of Foraminifera limestone (from the Middle and Upper Paleocene). The layers dip toward the southwest (250/20). There are no significant regional tectonic elements (faults, beds, etc.) in or near the quarry. The surrounding tectonic area is part of the Trieste–Komen plateau (see Figure 3).

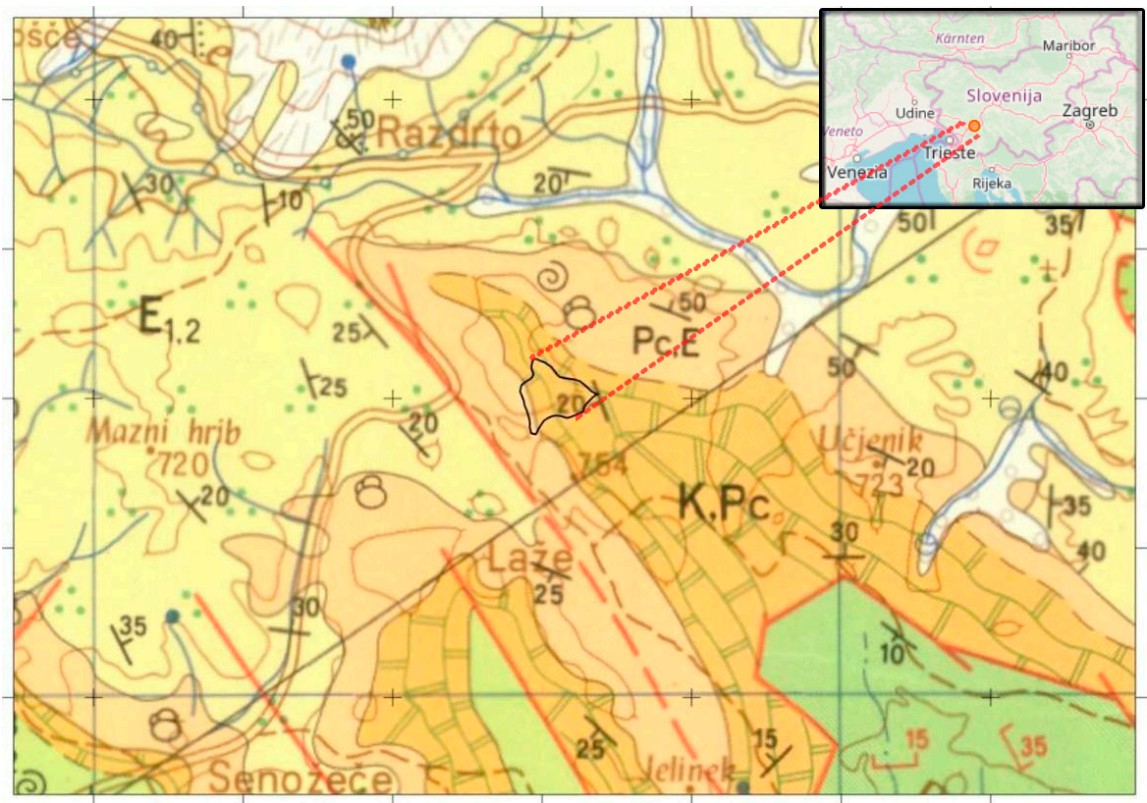

**Figure 3.** Location of the Laže quarry in the Official geological map of Slovenia (Postojna sheet) (public access on [60]).

## 2. Materials and Methods

### 2.1. Materials

#### 2.1.1. Digital Terrain Model

A digital terrain model was created using a 1 × 1 m grid to obtain a clear and authentic model. Several grids were created, which were combined with each other:

- A digitization of the original terrain of the topographic map M 1:5000 and construction of 5 × 5 m mesh;
- LIDAR terrain record data from 2014, including intensity images created from LIDAR recorded in the area. The data are available in the Environmental atlas of Slovenia [59]. A more precise (1 × 1 m) digital terrain model (DTM) was created;
- A digitalization of existing quarry products on a geodetic map. The map had been constantly updated by quarry management. Using the map from December 2018, we digitized the benches and created a digital model of the quarry. This model showed the benches before the collapse;
- A review and comparison of old aerial photographs as well as occasional quarry recordings; and
- Unmanned aerial photography data from 8 May 2019.

#### 2.1.2. Topography and Geomorphological Analysis of the Relief

We used various topographical maps and geomorphological analyses in the form of a slope map, a terrain orientation map, etc., to interpret the relief. Three-dimensional maps of relief features and colored orthophotos are upgrades of classical maps.

The surface around the quarry was analyzed to find the relationship between visible anomalies and discontinuities. The surface model was generated from aerial laser-scanning data (LIDAR 2014).

*2.2. Methods*

### 2.2.1. Sliding Surface

As is well-known, a photorealistic 3D model of a quarry is a combination of aerial imagery overlaid with a digital model of the terrain. The geometry of the 3D model is defined by a grid of irregular triangles (TIN grid). Using a photorealistic graphical background, a visible area of the sliding surface was easily pinpointed. By selecting this area on the network of triangles or on the 3D model, the exact sliding surface relief area was determined. The marked area of the 3D model was used as an input to calculate the spatial plane. The method that had the best fit (for the indicated relief) was used for the calculation. This plane represented an important discontinuity surface along which the rock mass slid.

### 2.2.2. Quarry Mapping

The purpose of the geological fieldwork was to identify the major joint sets in the rock mass and to classify the rock mass according to typical classification systems [61–63]. The joint roughness coefficient, which determines the shear strength of discontinuities, was measured with a profilometer (a Barton comb).

### 2.2.3. Stability Analysis

In order to assess the stability of the quarry slopes, both finite element methods (FEM) and the slice method were used. After a preliminary stability analysis of the entire quarry, a detailed analysis was then performed in the area affected by the March 2019 rockslide. In this way, it was possible to determine the significant features of the collapsed area in the Laže 1 quarry.

## 3. Results

*3.1. Digital Relief Model with a Photorealistic Raster Graphic Base*

On 8 May 2019, we performed an aero-photogrammetrical survey of the entire quarry surface using an unmanned aircraft. A total of 422 aerial photographs were taken. The photos had an 80% overlap with each other and a 70% overlap with the bands. The measurements were integrated into the D96/TM (ETRS 89) national coordinate system using the RTK GNSS method. Eight control points were measured with centimetric accuracy with a Leica GS18T receiver. A 10 × 10 cm grid was also constructed (see Figure 4).

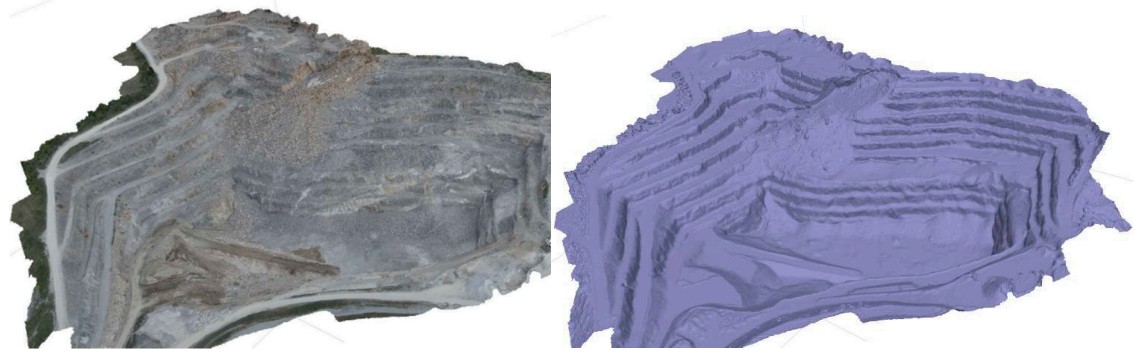

**Figure 4.** Laže 1 quarry, 3D view toward the north. The terrain model was created using unmanned aircraft photography. Left: the model is covered with photos. Right: a digital relief model with a 10 × 10 cm grid.

The main focus of the modeling was to find the overall spread of the main systems of cracks and fractures and to project these systems onto the unexcavated part of the quarry.

In carrying out each process, we searched for a model 3D image of the quarry and gradually implemented a model to verify it (both computationally and from an engineering perspective) and project it onto the rest of the quarry (see Figure 4).

In this way, the digitized data we created were integrated into the LIDAR terrain mesh. We obtained four products for use and interpretation (see Figure 5):

- a terrain model of the quarry before exploitation;
- a quarry model using 2014 LIDAR data;
- a model of the quarry from December 2018; and
- a quarry model with data from 8 May 2019, i.e., after the rockslide.

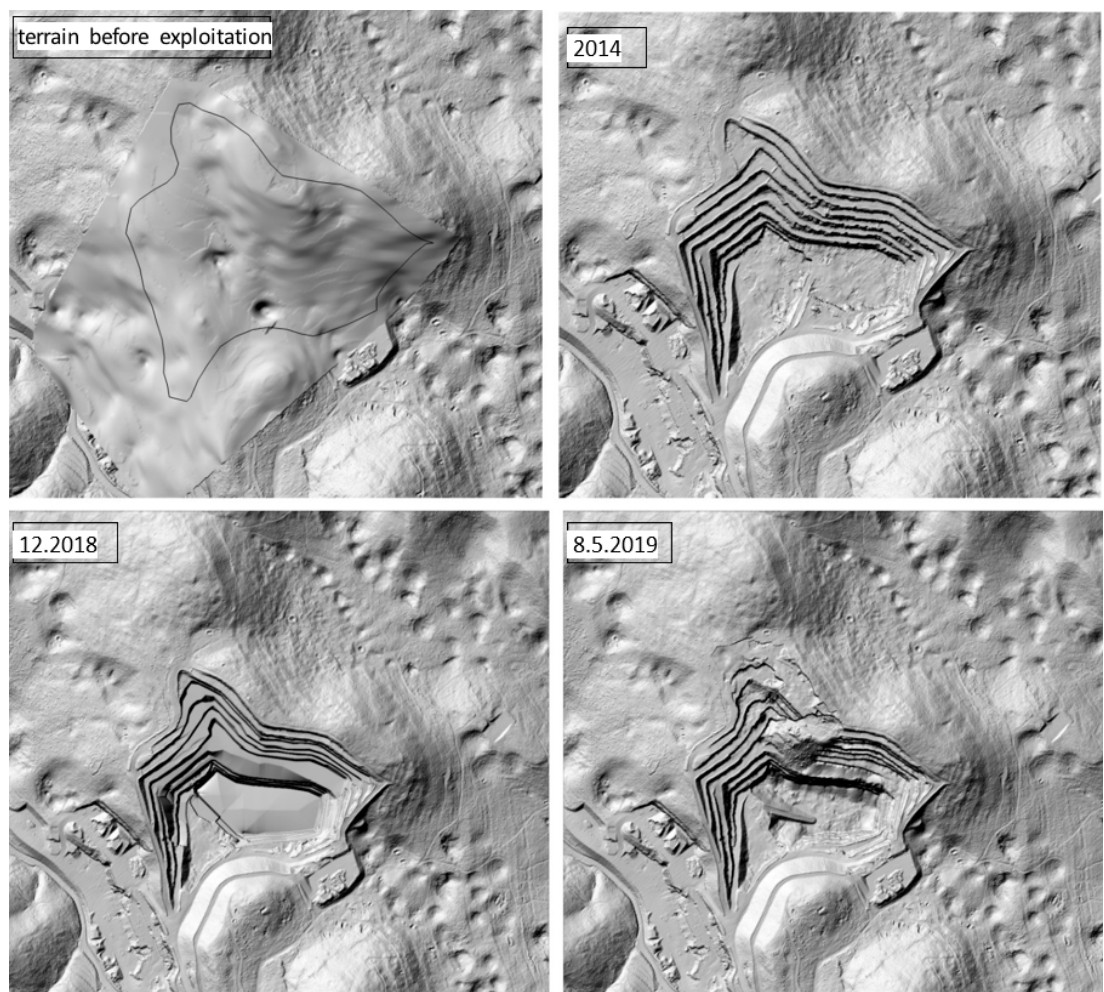

**Figure 5.** Shaded models of quarry relief in the four phases of the Laže 1 quarry operation.

These products were sufficient for what we needed for interpretations and projections onto the unexcavated part of the quarry.

On 13 March 2019, during a blasting operation, a rockslide of about 47,800 m$^3$ with a mean width of about 60 m occurred, affecting all quarry benches from the top bench to the bench at 700 a.s.l. The landslide debris slid down to the lower benches, reaching the quarry bottom (at about 672 m a.s.l.). A portion of the landslide debris stopped in the upper parts of the quarry. At the northern and eastern flanks of the excavated area, vertical cracks clearly delimited the landslide debris, whereas along the eastern and western flanks, the slopes, which had not yet collapsed, showed the presence of vertical cracks. A straight sliding surface was also clearly visible.

### 3.2. Mapping of Discontinuities

The joint systems affecting the rock mass were mapped and classified into several groups (according to their importance for slope stability) as follows (see Figure 6):

- Cracks facing southwest at 20°–40°. These included the major discontinuity due to the rockslide (245/21). This crack was perfectly straight and was the main crack (see Figure 2). This also included a crack that was parallel to the main crack (9.57 m), right above it. On the benches at 730 and 700 a.s.l., tension cracks spread from these benches to several other benches (234/23). In the eastern part of the quarry, the cracks were filled with calcite. Along these cracks, we noticed major sliding;
- Cracks perpendicular to the main crack system 40–60/60–65: these cracks occurred throughout the quarry and were clearly visible in the area of the rockslide;
- Vertical and subvertical cracks with a strike running from northeast–southwest;
- Other crack systems that did not belong to any of the above crack systems.

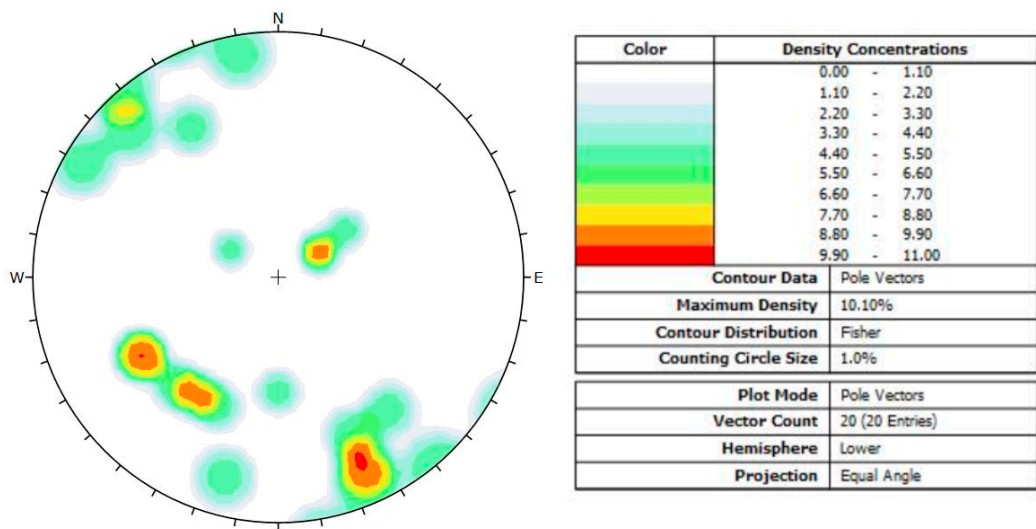

**Figure 6.** Schmidt diagram poles of cracks (discontinuities) in the Laže 1 quarry.

### 3.3. Structurally Connecting Tectonic Elements and Projections

During the quarry mapping, it was difficult to connect cracks throughout the quarry due to the inaccessibility of individual benches and the inability to check individual benches. For this purpose, a digital model was used to map existing cracks. These connected cracks were imaged using a structural map of the quarry, and we constructed a map of the plane and the intersection between the regular quarry surface and the sliding surface. A spatial representation of this process is shown in Figure 7.

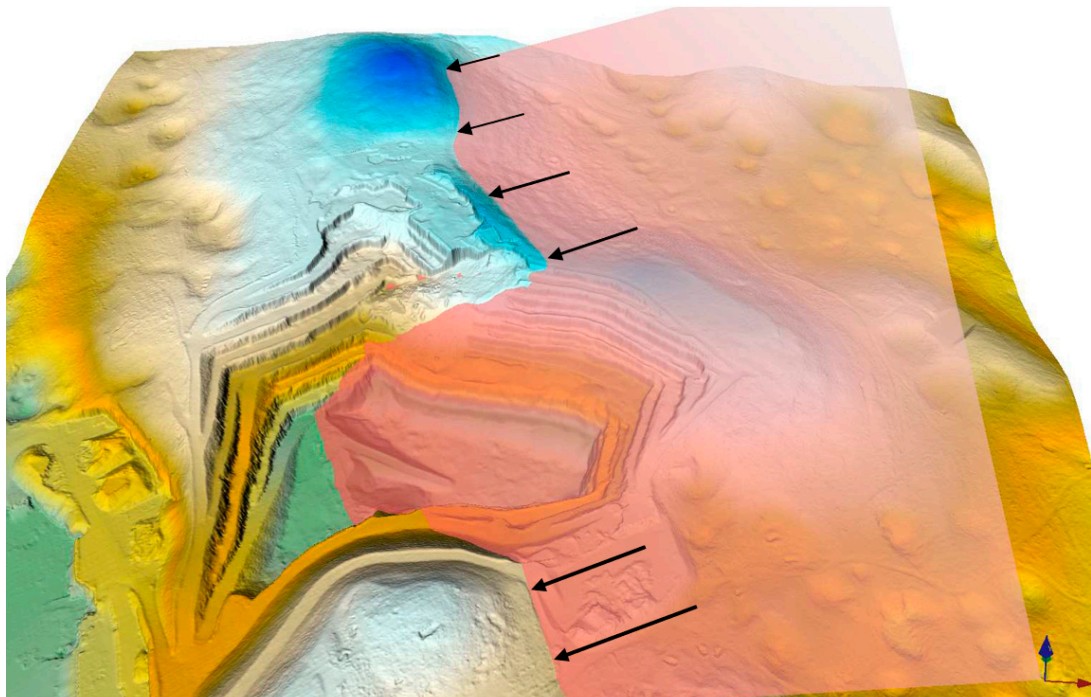

**Figure 7.** The Laže 1 quarry (a view to the north). Using the mapping data, the plane of the main crack (red surface) was constructed and put into a digital relief model. This gave us a spatially visible and determinable projection of the still-unexcavated areas and the entire surface (arrows).

The spatial orientation of a sliding surface is defined by the direction of the normal vector, which here was [−0.324150, −0.150404, 0.933973]. From this, the slope (21°) and azimuth (245°) of the sliding surface were calculated.

## 4. Discussion

The UAV aero-photogrammetric measurements were very useful for the interpretation of the discontinuities within the quarry. Using a photorealistic 3D model of the quarry surface, it was possible to locate discontinuities and cracks and determine the parameters of the rock for a stability analysis. The 3D model of the quarry and the digital model of the surrounding area gave us a good indication of potential problems.

### 4.1. Laže Quarry Stability Assessment: Crack Projections

Through the stability analysis, potential paths for the system of major cracks were identified: by major cracks, we mean those with a south–west incidence and a low angle. After detailed research that included connecting benches and a visual assessment of the cracks in the Laže 1 quarry, five cracks were identified (Figure 8). We numbered them from west to east:

- A crack that was visible in the collapsed area in the vertical wall was parallel to the crack that led to the rockslide. It was 9.72 m away (height-wise) and is indicated as Number 1 (pictured in yellow) below;
- The crack that caused the collapse was by far the most susceptible to sliding. It was completely flat and extended over a large portion of the open quarry. We refer to it here as the main crack (Number 2, shown in red). This crack was clearly visible in surface anomalies;
- On the benches at 720 and 710 a.s.l., an intense crack from the same system was observed. It presented over several benches (234/23) and was clearly parallel to cracks a1 and a2 (Number 3, pictured in blue);

- A steeper crack that ran across multiple benches (235/37, pictured in orange, Number 4);
- In the eastern part of the quarry, on bench 680 (a.s.l.), more calcite-filled cracks were found (230/38). Along these cracks, the banks of the benches were falling off. The projections were dragged along the benches for analysis (pictured in green, Number 5).

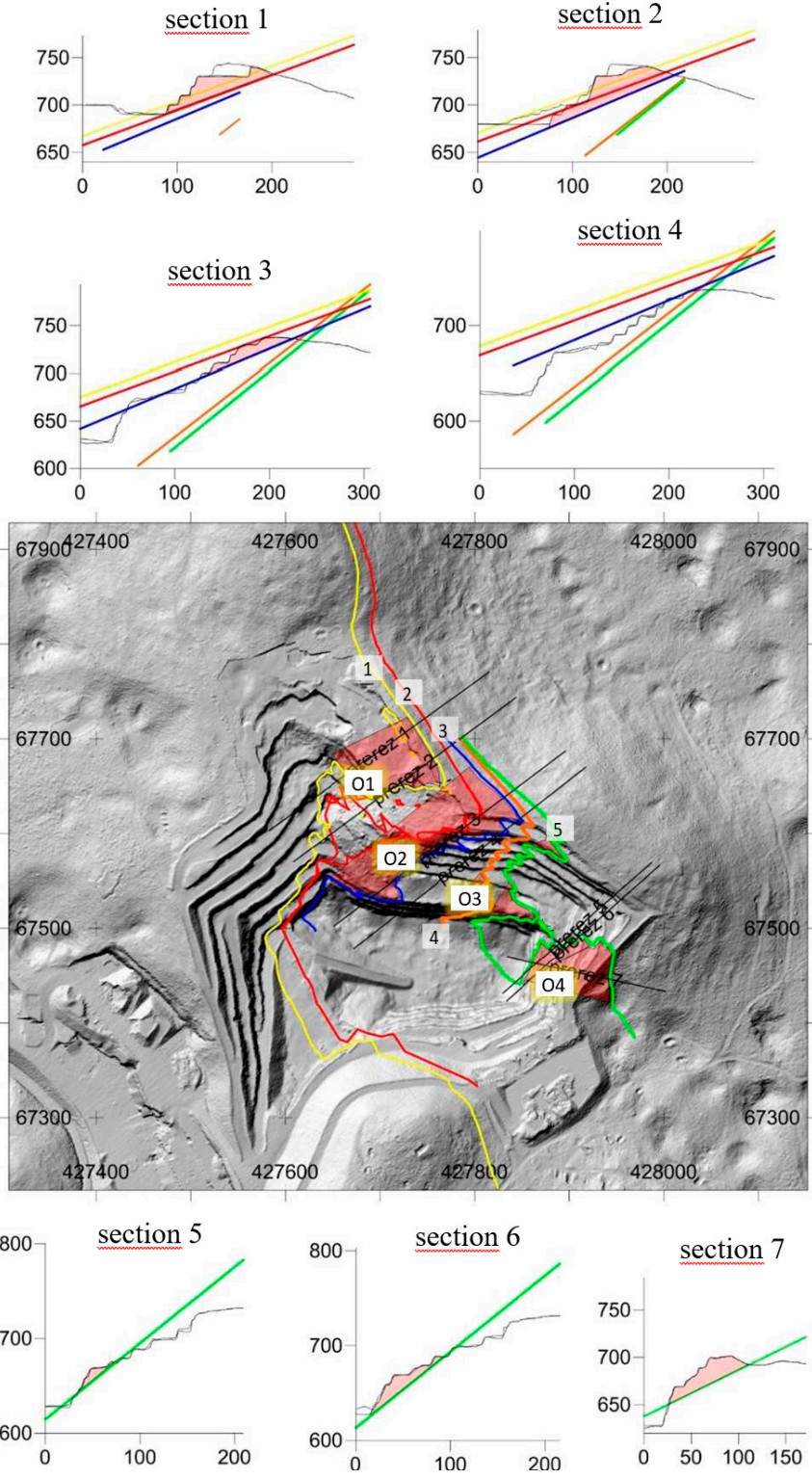

**Figure 8.** The locations of major discontinuities (numbers 1 to 5) and critical areas in the quarry (numbers O1–O4) are marked in red.

All of these cracks were projected onto the slopes (the quarry benches). We were looking for free paths, or rather, a situation that was similar or identical to the situation prior to the March 2019 slide: a faulty crack on the slope combined with a vertical crack, forming a structural wedge. Therefore, four areas were determined to be critical areas:

- The first area, in the western part of the quarry, was due to cracks 1 and 2. One slide had already occurred at crack number 2, but the cracks were clearly still spreading westward in such a way that a similar situation could occur again. The likelihood of collapse is high;
- The second area was at crack number 3. This crack had the same free path as the first two cracks;
- The third area was by crack number 4, which had a larger vertical drop than the first three cracks did. The vertical crack, which was observed to run through all of the quarry benches on this slope, created a wedge between benches 680 and 622 (a.s.l.);
- The fourth area, which was in the eastern part of the quarry, had the same combination of cracks as did crack number 5 in area number 3.

### 4.2. Morphography and Surface Morphology

A view of the terrain before exploitation indicated two prominent sinkholes that were previously connected by tectonic lines. The sinkholes are oriented NW–SE due to tectonics or due to the orientation of the layers and weathering along the cracks parallel to the layers.

Figure 9 shows the terrain surrounding the Laže 1 quarry, which was constructed from aerial laser scanning data (LIDAR 2014). The red arrows indicate an anomaly on the surface due to a connection to the crack that led to the collapse.

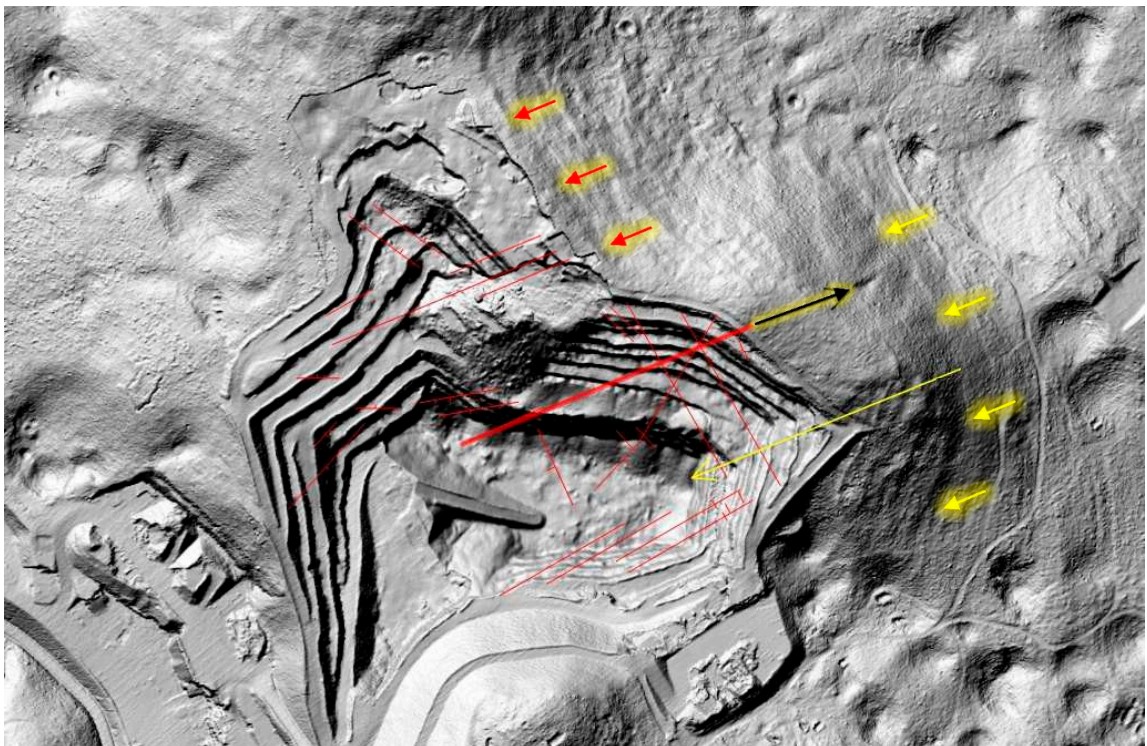

**Figure 9.** The morphological peculiarities of the quarry were also reflected on the surface: the red arrows indicate a groove in the surface, which indicates a connection with the crack where the collapse took place. The yellow arrows mark recesses, which are spread throughout quarry due to plane construction. The vertical fracture in the quarry is expressed through a morphological irregularity on the surface (black arrow).

The yellow arrows mark irregularities, which spread throughout the quarry due to plane construction. The vertical fracture is expressed through a morphological groove on the surface (black arrow).

Recording the terrain with methods that allow for high accuracy (LIDAR, photogrammetry, etc.) revealed surface characteristics that could be connected after an inspection of the structural tectonic elements of the quarry was done. In this way, tectonic elements could be projected onto nonexcavated parts of the areas being exploited.

## 5. Conclusions

This paper covers a rockslide that occurred in March 2019 in the Laže 1 quarry and the resulting instability. The rockslide happened due to a pronounced flat crack in the layers and an unfavorable incidence of this crack relative to the front slope. Similar situations may occur during further excavation, which should be considered in the design and execution of the excavation.

Based on the calculations and good engineering judgment, the rocks in the Laže 1 quarry are generally stable. The rocks on the front slopes are more damaged due to the impact of blasting. This can lead to local instability, which is related to local discontinuities.

Correct interpretations of the morphological forms in a quarry and its surroundings are very important in examining unexcavated areas. Modern measurement technologies can provide accurate digital terrain models and enable the collection of quality data for surface stability analyses. Here, we were able to perform a stability analysis by combining aero-photogrammetric data and LIDAR data to discover discontinuities, cracks, and the parameters of the rock. Some morphologic anomalies in the surfaces of the nonexcavated areas were connected to discontinuities that were visible in the opened quarry using precise surveying and proper interpretation.

**Author Contributions:** Conceptualization, A.L. and M.V.; Formal analysis, A.L., T.B., and G.V.; Methodology, M.V. and T.B.; Writing—review and editing, A.L. and T.B. All authors have read and agreed to the published version of the manuscript.

**Funding:** The APC was funded by the Faculty of Natural Sciences and Engineering.

**Acknowledgments:** We would like to thank the companies Kolektor CPG d.o.o., Geotrias d.o.o., and Geoservis, d.o.o., for providing us access to the collected data. This research was supported by the Slovenian Ministry of Education, Science, and Sport; the Faculty of Natural Sciences and Engineering; and research program P2-0268, financed by the Slovenian Research Agency. We would also like to thank the anonymous reviewers and members of the editorial team for their comments.

**Conflicts of Interest:** The authors declare no conflicts of interest.

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
