# Peer review of "The Use of Precise Survey Techniques to Find the Connection between Discontinuities and Surface Morphologic Features in the Laže Quarry in Slovenia"

_minerals, doi:10.3390/min10040326_

Round 1
Reviewer 1 Report
Many of the concepts introduced in this paper are already well known to the scientific community. However, this paper illustrates an interesting case study which is worth publishing. My last consideration is that there are many pioneering works on the use of photogrammetry for rock mechanics surveys and discontinuity identification which are not referenced in your paper. These works have, in the reviewer opinion, much larger conceptual significance than those presenting the mere application of new technical instruments (such as UAV's) to obtain the results. This is the reason why I think that the references could be improved.
Author Response
Dear reviewer,
many thanks for your constructive reviews of submitted manuscript. We considered all your comments and suggestions for corrections and corrected all mistakes you have noticed. We believe that this has led to an improved and more systematic and concise manuscript. The responses to your comments are marked red.
Point 1: 1/ +: Many of the concepts introduced in this paper are already well known to the scientific community. However, this paper illustrates an interesting case study which is worth publishing. My last consideration is that there are many pioneering works on the use of photogrammetry for rock mechanics surveys and discontinuity identification which are not referenced in your paper. These works have, in the reviewer opinion, much larger conceptual significance than those presenting the mere application of new technical instruments (such as UAV's) to obtain the results. This is the reason why I think that the references could be improved.
Response 1: Your suggestion has been taken into account. New references were added to the manuscript.

Reviewer 2 Report
Dear Editor, Authors tried to monitor a slope stability problem in a stone quarry using UAV and analyze the stability of the some slopes depending on the movement. Stability assessment is not an easy work. Behaviour of the rock slopes must be totally understood. For example in this study area, there are planar failures exist depending on main fault and its discontinuities. Authors did the stability analyses and found circular surface in their study. This is not possible. Mapping discontinuities using remote sensing techniques is very innovative but researcher must know which discontinuity sets play an active role in stability problems. Engineering geologists just can do this kind of work. Without them, authors should not mention about and slope stability analyses. Monitoring land movements with UAV is enough for the research. I suggest major revision for the manuscript and English is very poor.
Regards,
Author Response
Dear reviewer,
many thanks for your constructive reviews of submitted manuscript. We considered all your comments and suggestions for corrections and corrected all mistakes you have noticed. We believe that this has led to an improved and more systematic and concise manuscript. The responses to your comments are marked red.
Point 1: 1/ +: Authors tried to monitor a slope stability problem in a stone quarry using UAV and analyze the stability of the some slopes depending on the movement. Stability assessment is not an easy work. Behaviour of the rock slopes must be totally understood. For example in this study area, there are planar failures exist depending on main fault and its discontinuities. Authors did the stability analyses and found circular surface in their study. This is not possible. Mapping discontinuities using remote sensing techniques is very innovative but researcher must know which discontinuity sets play an active role in stability problems. Engineering geologists just can do this kind of work. Without them, authors should not mention about and slope stability analyses. Monitoring land movements with UAV is enough for the research. I suggest major revision for the manuscript and English is very poor.
Response 1: Your suggestion has been taken into account. English language in article was improved.
The main topic of the article was to find connection between discontinuities, visible in the quarry and in surface visible irregularities in terrain. Mapping with UAV and use of photogrammetry was very useful tool for that identification.
The stability analysis, we used, consisted of several analyses, which examined together give proper answer of stability of the quarry. Procedure is described in major textbooks (e.g. Wyllie and Mah. Rock slope engineering, 4th edition 2001, pg. 231). First step in analysis is rock mass failure identification. Stability analysis was done by engineering geologist. For more clear illustration of main topic of the article stability analysis was removed from article, as suggested by all reviewers.

Reviewer 3 Report
This is a new version of a manuscript I already revised. My previous main objection concerned the validity of the numerical stability analysis based on curvilinear surfaces. The Authors accepted my suggestion and deleted this unnecessary and wrong analysis. Now, the manuscript is more consistent with the goals the Authors wished to achieve by means of a digital terrain model successively integrated by field surveys and unmanned aerial vehicles. In my opinion, the job retains its modest interest in reading for engineering geologists but it is more correct in its initial assumptions and consequential developments. The inclusion of suitable references and comments tempered the impression that this was only a technical report by practitioners. However, a further effort concerning the revision of the English grammatical form yet deficient must be carried out. I tried to improve several sentences and paragraphs but I am not an English mother-tongue (see the attached pdf). Furthermore, sometimes the text is not yet clear in several parts needing a profitable collaboration between the Authors and the translator. At last, I suggest combining in a single paragraph Discussion and Conclusions. Now, the manuscript can be accepted with minor corrections.

Author Response
Dear reviewer,
many thanks for your constructive reviews of submitted manuscript. We considered all your comments and suggestions for corrections and corrected all mistakes you have noticed. We believe that this has led to an improved and more systematic and concise manuscript. The responses to your comments are marked red.
Point 1: 1/+: This is a new version of a manuscript I already revised. My previous main objection concerned the validity of the numerical stability analysis based on curvilinear surfaces. The Authors accepted my suggestion and deleted this unnecessary and wrong analysis. Now, the manuscript is more consistent with the goals the Authors wished to achieve by means of a digital terrain model successively integrated by field surveys and unmanned aerial vehicles. In my opinion, the job retains its modest interest in reading for engineering geologists but it is more correct in its initial assumptions and consequential developments. The inclusion of suitable references and comments tempered the impression that this was only a technical report by practitioners. However, a further effort concerning the revision of the English grammatical form yet deficient must be carried out. I tried to improve several sentences and paragraphs but I am not an English mother-tongue (see the attached pdf). Furthermore, sometimes the text is not yet clear in several parts needing a profitable collaboration between the Authors and the translator.
Response 1: Your suggestion has been taken into account. English language in article was improved. The text was corrected as suggested.
Point 2: 1/+: At last, I suggest combining in a single paragraph Discussion and Conclusions. Now, the manuscript can be accepted with minor corrections.
Response 2: The modern form of article structure in Minerals requires separate chapters for Discussion and Conclusion. Other reviewers had no comment on the structure, so we decided to keep original form.
Point 3: 9/ 237 line: Where are the arrows?
Response 3: Figure 7 is corrected.

Round 2
Reviewer 2 Report
I strictly advised the removing the stability parts of the manuscript. Authors removed one figure but the sections are still exist.
Nobody can do stability analysis without using some engineering parameters such as unit weight, cohesion, internal friction angle etc.
Without these, using stereographic projection, just kinematic analysis could be done.
I understand from what they wrote in the manuscript, there is no expert in their team about kinematic analysis and slope stability.
I suggest again, giving just slope monitoring data obtained from UAV is enough.
I suggest major revision again.
Author Response
Dear reviewer,
many thanks for your reviews of submitted manuscript. We considered all your comments and suggestions for corrections and corrected you have noticed. We believe that this has led to an improved and more systematic and concise manuscript. The responses to your comments are marked red.
Point 1: 1/ +: I strictly advised the removing the stability parts of the manuscript. Authors removed one figure but the sections are still exist.
Response 1: Your suggestion has been taken into account. The text was corrected as suggested.
Point 2: 1/ +: I suggest again, giving just slope monitoring data obtained from UAV is enough.
Response 2: Your suggestion has been taken into account. We have removed the chapters ”2.2.3. Stability analysis” and “3.4. Stability of the Laže quarry”. Thank you for your comment.
Yours sincerely,
Aleš Lazar, corresponding author

Round 3
Reviewer 2 Report
Dear Authors,
Please check English of the whole manuscript with the native speaker.
This manuscript is a resubmission of an earlier submission. The following is a list of the peer review reports and author responses from that submission.
Round 1
Reviewer 1 Report
Please see my comment on the attached file

Reviewer 2 Report
My some brief comments are summarized below;
Page 2, Line 45-51, do not use “we”, all statements must be passive.
Line 62, delete “and”
Figure 2,3 : If there are two figures placed, so there must be separated to figure captions below the figures. Otherwise, there is a figure prepared with a one figure number and caption. Correct figure number or caption.
In Figure 2.3, The flat surface is a fault plane, authors must emphasize and consider this surface during their stability analysis.
Page 3, Line 74. Authors mention about basic geological map. Who prepared this map? If it is previously prepared so they must refer it.
Page 4, Line 92, authors mention about photos. What kind of photos they refer to?
All “we”s must be removed in the manuscript.
Page 5, Line 111, what does “evenness” mean? This word is not suitable for discontinuities. Smoothness could be preferred.
Page 5, Line 110, Sclerometer is not used in these kind of works. Schmidt hammer could be used.
Figure 11, drawing all great circles belonging the discontinuity sets is meaningless and must be deleted. Great circles belonging to the main pole concentration points could be drawn.
Page 6-7, authors tried to group the discontinuity sets but the method is unclear. Discontinuitie are grouped according to the Schmidt nets.
Figure 12, authors are giving a failure surface overlapped on their digital 3D model but the failure surface is wrong. It is not controlling their slide. If they look to their field photo. Their slide is very small.
Page 11, Figure ….., two figures and must be corrected as one!
SRF value is higher that 1.25 which is 2.18. This means that the analysed slope is stable but their slope is sliding. How authors explain this?
Table 5, replace “incidence” with “dip angle”
Figure 16. What does “groove” mean? Do authors try to say tension crack?
Reviewer 3 Report
This manuscript presents a stability analysis of several benches belonging to a quarry located in Slovenia, and affected in the past by a rockslide. In order to perform this analysis a digital terrain model was prepared and successively integrated by discontinuity surveys and geological mapping. Then, the recognized slope failure model was geothecnically and geomechanically analysed using both FEM method and geostructural approaches based on the main joint sets.
While the text is relatively easy to read, my major objection concerns the novelty of the paper. It seems more like a technical report by practitioners than a scientific paper. What is new in this paper? What potential readers can learn from this paper? It uses means and methods which have long been used in the practical approaches concerning the rock slope stability. Furthermore, I don’t understand why the Authors use a failure model based on a curvilinear surface (better suited for soils) while they identify a planar discontinuity surface as responsible for the whole rockmass failure. What is the analysed model which in a better manner simulates the actual situation? The document is missing precise details concerning the Discussion and only a few References are quoted in the text although a long list appears in it. At last, a careful review by an English mother-tongue is need.
Due to the above remarks, I think that the document cannot be accepted in this high impact Journal and it must be rejected.